# Some Recent Advances in Energetic Variational Approaches

**DOI:** 10.3390/e24050721

**Published:** 2022-05-18

**Authors:** Yiwei Wang, Chun Liu

**Affiliations:** Department of Applied Mathematics, Illinois Institute of Technology, Chicago, IL 60616, USA; ywang487@iit.edu

**Keywords:** energetic variational approach, non-equilibrium thermodynamics, chemo-mechanical coupling, thermal effects

## Abstract

In this paper, we summarize some recent advances related to the energetic variational approach (EnVarA), a general variational framework of building thermodynamically consistent models for complex fluids, by some examples. Particular focus will be placed on how to model systems involving chemo-mechanical couplings and non-isothermal effects.

## 1. Introduction

Complex fluids comprise a large class of soft materials, such as polymeric solutions, liquid crystals, ionic solutions, and fiber suspensions. These are fluids with complicated rheological phenomena, arising from different “elastic” effects, such as the elasticity of deformable particles, interaction between charged ions, and bulk elasticity endowed by polymer molecules [1,2]. Due to their strong nonlinear and non-equilibrium nature, building thermodynamically consistent models for complex fluids has been an interesting and challenging problem. The difficulty arises from complicated coupling and competition of different electro-chemo-mechanic mechanisms, such as long-range interaction and thermal fluctuation, in different spatio-temporal scales.

Motivated by the non-equilibrium thermodynamics, especially the seminal works of Rayleigh [3] and Onsager [4,5], the energetic variational approach (EnVarA) has proven to be a powerful tool in studying numerous complex fluids systems in physics, chemistry, and biochemistry [6,7], including liquid crystals [8,9], viscoelastic fluids [10], multiple-phase flows [11,12], ionic solutions [13], etc. The idea of EnVarA is to describe a complex system by an energy–dissipation law
(1)$$\frac{d}{dt}E^{\mathrm{total}}=-▵\leq 0,$$
where $E^{\mathrm{total}}$ is the sum of the kinetic energy K and the Helmholtz free energy F, ▵ is the rate of energy dissipation. The energy–dissipation law (Equation 1) can be obtained by combining the first and second laws of thermodynamics for an isothermal and mechanically isolated system [14]. Indeed, the first law of thermodynamics states that
(2)$$\frac{d}{dt}\left(K+U\right)=\dot{W}+\dot{Q},$$
where K is the kinetic energy, U is the internal energy, $\dot{Q}$ is the rate at which heat absorbed from the environment, and $\dot{W}$ is the rate of the external work done by the environment. To analyze heat, one introduces the second law of thermodynamics
(3)$$T\frac{dS}{dt}=\dot{Q}+▵,$$
where *T* is the absolute temperature, *S* is the entropy, and ▵≥0 is the rate of entropy production. By subtracting the first law (Equation 2) and the second law (Equation 3), we obtain an energy–dissipation law (Equation 1) for an isothermal ($\frac{d}{dt}T=0$) and mechanically isolated ($\dot{W}=0$) system with F=U−TS being the Helmholtz free energy. So, in this case, the rate of energy dissipation equals the rate of entropy production.

Starting with an energy–dissipation law (Equation 1), EnVarA derives the dynamics of the systems through the least action principle (LAP) and the maximum dissipation principle (MDP). The LAP, which states the equation of motion for a Hamiltonian system can be derived by taking variation of the action functional $A=\int_{0}^{T}K-Fdt$ with respect to the trajectory x, gives a unique procedure to derive the conservative force in the system. The MDP, variation of the dissipation potential D with respect to $x_{t}$, i.e., the velocity, derives the dissipation force in the system. In turn, the force balance condition leads to the underlying evolution equation of the system
(4)$$\frac{\delta D}{\delta x_{t}}=\frac{\delta A}{\delta x}.$$ According to the Onsager theory [4,5], the rate of entropy production ▵ is quadratic in terms of $x_{t}$ in the linear response regime, and the dissipation potential $D=\frac{1}{2}▵$ in this case.

There are other forms of variational principles, such as the general equation for non-equilibrium reversible–irreversible coupling (GENERIC) [1,15,16], Onsager’s variational principle [17,18,19,20], and conservation–dissipation formalism [21,22], that have also been helpful in studying complex fluids. Although these variational principles are equivalent to EnVarA in most cases, these approaches are based on the principle of virtual work (PVW), in which the variation is often taken with respect to the state variables directly. The state variables are defined in Eulerian coordinates in many cases, and can be viewed as generalized coordinates in these systems [17].

In contract to other variable principles, the classical EnVarA formulation is rooted in the continuum mechanics [23,24,25] and a Lagrangian formulation of an underlying system. Continuum mechanics is a generalization of Newtonian particle mechanics. In continuum mechanics, it is assumed that there are infinite many particles form a continuum body. The motion of these particles is described by a flow map x(X,t), where X is the Lagrangian coordinate and x is the Eulerian coordinate. More importantly, in the context of continuum mechanics, all dynamics of the employed variables are determined by the flow map x(X,t) and its derivatives (the velocity and the deformation tensor) through their kinematics. The energy–dissipation law, together with the kinematics of the employed variables, describes all the physics and assumptions in the system. It is important to realize that systems derived by the LAP can also be derived by the PVW, but not vice versa. In fact, the discrepancy between the LAP the PVW is of great interest in the mathematical theory of weak solutions and the theory of singularities [8,11].

The purpose of this paper is to provide a brief overview of some recent developments of the energetic variational approach, especially for the systems involving chemo-mechanical couplings and non-isothermal effects, through examples. We refer interested readers to [6,7] for more details of the classical energetic variational approach. The rest of the paper is organized as follows. In Section 2, we briefly review the EnVarA in continuum mechanics, and show its applications in modeling generalized diffusion, a dilute polymeric fluids, and the kinetic Fokker–Planck equation. The EnVarA formulation for reaction kinetics and its applications in modeling chemo-mechanical systems, such as reaction–diffusion systems, the dynamical boundary conditions, the Boltzmann equation and the reaction/active fluids, are given in Section 3. In Section 4, we discuss the extension of the EnVarA to non-isothermal systems. Some applications of the EnVarA in developing structure-preserving numerical schemes and thermodynamically consistent coarse-grained model are reviewed in Section 5.

## 2. EnVarA in Continuum Mechanics

As mentioned in the introduction, the classical energetic variational approaches are variational principles for continuum mechanics [7], and the variable x in (Equation 4) should be understood as the flow map x(X,t) from a reference domain $\Omega_{0}$ to a physical domain $\Omega_{t}$. Here $X\in \Omega_{0}$ is the Lagrangian coordinate and $x\in \Omega_{t}$ is the Eulerian coordinate. For a fixed X, x(X,t) describes a trajectory of a particle (or a material point) labeled by X, while for a fixed *t*, x(X,t) is a diffeomorphism from $\Omega_{0}$ to $\Omega_{t}$ (see Figure 1 for an illustration).

An important feature of a continuum mechanical system is that the evolution of physical variables, such as the density function, are determined by the evolution of the flow map x(X,t) through kinematics. All the kinematic transport information of these variables is carried by the deformation tensor F. For a given flow map x(X,t), the deformation tensor F is defined by
(5)$$\tilde{F}\left(x\left(X,t\right),t\right)=F\left(X,t\right)=\nabla_{X}x\left(X,t\right).$$ A direct computation shows that $\tilde{F}$ satisfies a transport equation [7]
$${\tilde{F}}_{t}+u\cdot \nabla \tilde{F}=\nabla u\tilde{F}.$$ Without ambiguity, we will not distinguish F and $\tilde{F}$ throughout this paper. Due to the conservation of mass, $\int_{\Omega_{t}}\rho \left(x,t\right)dx=\int_{\Omega_{0}}\rho_{0}\left(X\right)dX$, the kinematics of a density function ρ(x,t) can be written as
(6)$$\rho \left(x,t\right)=\rho_{0}\left(X\right)/\det F\left(X,t\right)$$
in Lagrangian coordinates, where $\rho_{0}\left(X\right)$ is the initial density. The kinematics (Equation 6) is equivalent to the continuity equation $\rho_{t}+\nabla \cdot \left(\rho u\right)=0$ in Eulerian coordinates. For a scalar variable that is purely transported by the flow map, the kinematics is given by
$$\varphi \left(x\left(X,t\right),t\right)=\varphi_{0}\left(X\right)\;\mathrm{or}\;\varphi_{t}+\nabla \varphi \cdot u=0$$
in Eulerian coordinates. For a nematic liquid crystal that consists of rod–like molecule, the kinematics of the nematic order parameter $d\in R^{d}$ can be written as
$$d\left(x\left(X,t\right),t\right)=Fd_{0}\left(X\right)\;\mathrm{or}\;\partial_{t}d+\left(u\cdot \nabla \right)d-\left(\nabla u\right)d=0.$$

### 2.1. Generalized Diffusion

One of the simplest classes of mechanical processes is generalized diffusion. Generalized diffusion is concerned with a conserved quantity ρ(x,t) satisfying the kinematics
(7)$$\partial_{t}\rho +\nabla \cdot \left(\rho u\right)=0,$$
where u is the average velocity. In the framework of EnVarA, a generalized diffusion can be described an energy–dissipation law
(8)$$\frac{d}{dt}F\left[\rho \right]=-\int \eta \left(\rho \right){\left|u\right|}^{2}\;dx,\;F\left[\rho \right]=\int \omega \left(\rho \right)\;dx,$$
where ω(ρ) is the free energy density, η(ρ)>0 is the friction coefficient. Due to the kinematics (Equation 7), the free energy can be reformulated as a functional of x(X,t) in Lagrangian coordinates. A direct computation shows that
$$\begin{array}{cc} \; & \delta A=-\delta \int_{0}^{T}\int_{\Omega_{0}}\omega \left(\rho_{0}\left(X\right)/\det F\right)\det F\;dXdt\\ \; & =-\int_{0}^{T}\int_{\Omega_{0}}\left(-\frac{\partial \omega }{\partial \rho }\left(\frac{\rho_{0}\left(X\right)}{\det F}\right)\cdot \frac{\rho_{0}\left(X\right)}{\det F}+\omega \left(\frac{\rho_{0}\left(X\right)}{\det F}\right)\right)\times \left(F^{-T}:\nabla_{X}\delta x\right)\det F\;\;dXdt, \end{array}$$
where δx is the test function satisfying δx·n=0 with n being the outer normal of Ω in Eulerian coordinates (Here we will not distinguish $\tilde{\delta x}\left(x\left(X,t\right),t\right)=\delta x\left(X,t\right)$ and δx(X,t) without ambiguity). Pushing forward to Eulerian coordinates, we have
(9)$$\begin{array}{cc} \delta A & =-\int_{0}^{T}\int_{\Omega }\left(-\frac{\partial \omega }{\partial \rho }\rho +\omega \right)\nabla \cdot \left(\delta x\right)dxdt=\int_{0}^{T}\int_{\Omega }-\nabla \left(\frac{\partial \omega }{\partial \rho }\rho -\omega \right)\cdot \delta xdxdt, \end{array}$$
which indicates that
$$\frac{\delta A}{\delta x}=-\nabla \left(\frac{\partial \omega }{\partial \rho }\rho -\omega \right)=-\rho \nabla \mu ,$$
where $\mu =\frac{\delta F}{\delta \rho }$ is the chemical potential. In the notion of the principle of virtual work [24], one can obtain (Equation 9) by using the relation δρ=∇·(ρδx)

For the dissipation part, since $D=\frac{1}{2}\int \eta \left(\rho \right){\left|u\right|}^{2}dx$, it is easy to compute that $\frac{\delta D}{\delta u}=\eta \left(\rho \right)u$. As a consequence, we have the force balance equation
(10)η(ρ)u=−ρ∇μ. Combining the force balance Equation (Equation 10) with the kinematics (Equation 7), one can obtain a generalized diffusion equation
(11)$$\rho_{t}=\nabla \cdot \left(\frac{\rho^{2}}{\eta (\rho )}\nabla \mu \right).$$ It is worth mentioning that the above derivation is rather formal. A certain analysis is needed to show the existence of the flow map x(X,t). We refer interested readers to [26,27] for some related discussions.

Many classical models can be viewed as generalized diffusions with different forms of the free energy F[ρ] and the dissipation. For example, the porous medium equation (PME) $\rho =\Delta \rho^{m}$ can be obtained by taking $F\left[\rho \right]=\int \frac{1}{m-1}\rho^{m}dx$ and η(ρ)=ρ, the Cahn–Hilliard equations $\varphi_{t}=\Delta \left(-\Delta \varphi +f^{\prime }\left(\varphi \right)\right)$ can be obtained by taking $F\left[\varphi \right]=\int f\left(\varphi \right)+\frac{1}{2}{\left|\nabla \varphi \right|}^{2}dx$ and $\eta \left(\varphi \right)=\varphi^{2}$ [28], and the Poisson–Nernst–Planck (PNP) equations for ion transport
$$\left\{\begin{array}{cc} \; & n_{t}=\Delta n-\nabla \cdot \left(\nabla \phi n\right),\\ \; & p_{t}=\Delta p+\nabla \cdot \left(\nabla \phi p\right),\\ \; & \nabla (\epsilon \nabla \phi )=n-p. \end{array}\right.$$
can be obtained from the energy–dissipation law
(12)$$\frac{d}{dt}\int n\ln n+p\ln p+\frac{\epsilon }{2}{\left|\nabla \phi \right|}^{2}dx=-\int n|u_{n}{|}^{2}+p{\left|u_{p}\right|}^{2}dx,$$
along with the Poisson equation ∇(ϵ∇ϕ)=n−p [13].

Another interesting example of generalized diffusions is the nonlinear Fokker–Planck equation associated with a random process
(13)$$dX_{t}=a\left(X_{t}\right)dt+\sigma \left(X_{t}\right)dW_{t}.$$ It is well known that different definitions of stochastic integrals can lead to different diffusion equations. For the stochastic process (Equation 13), Itô calculus leads to a diffusion equation
$$\rho_{t}+\nabla \cdot \left(a\rho \right)=\frac{1}{2}\Delta \left(\sigma^{2}\rho \right),$$
which corresponds to the energy–dissipation law
$$\frac{d}{dt}\int \rho \ln\left(\sigma^{2}\rho /2\right)+\psi \rho dx=-\int \frac{\rho }{\sigma^{2}/2}{\left|u\right|}^{2}dx$$
if $a=\frac{1}{2}\sigma^{2}\nabla \psi$ [29]. Meanwhile, the Stratonovich integral yields
$$\rho_{t}+\nabla \cdot \left(a\rho \right)=\frac{1}{2}\nabla \cdot \left(\sigma \nabla \left(\sigma \rho \right)\right),$$
which corresponds to the energy–dissipation law
$$\frac{d}{dt}\int \rho \ln\left(\sigma f\rho \right)+\psi \rho dx=-\int \frac{f}{\sigma^{2}/2}{\left|u\right|}^{2}dx$$
if $a=\frac{1}{2}\sigma^{2}\nabla \psi$. In both cases, the condition $a=\frac{1}{2}\sigma^{2}\nabla \psi$, known as the fluctuation–dissipation theorem in statistical physics [30], is crucial for the existence of the energy–dissipation law, as well as the existence of an equilibrium state [7].

In the case that η(ρ)=ρ, generalized diffusions can be viewed as Wasserstein gradient flows in space of all probability densities having finite second moments $P_{2}\left(\Omega \right)$ [31]. Formally, the Wasserstein gradient flow can be defined as a continuous time limit (τ→0) of the semi-discrete scheme, known as the JKO scheme,
(14)$$\rho^{k+1}=\underset{\rho \in P_{2}\left(\Omega \right)}{\arg\min}\frac{1}{2\tau }W_{2}\left(\rho ,\rho^{k}\right)+F\left[\rho \right],\;k=0,1,2\ldots ,$$
where $P_{2}\left(\Omega \right)=\{\rho :\Omega \to \left[0,\infty \right)\;|\;\int_{\Omega }\rho dx=0,\int_{\Omega }{\left|x\right|}^{2}\rho \left(x\right)dx<\infty \}$ and $W_{2}\left(\rho ,\rho^{k}\right)$ is the Wasserstein distance between ρ and $\rho^{k}$. The Wasserstein distance between two probability densities $\rho_{1}$ and $\rho_{2}$ can be computed through a Benamou–Brenier formulation [32]
(15)$$W_{2}\left(\rho_{1},\rho_{2}\right)=\underset{(\rho ,u)\in S}{\arg\min}\int_{0}^{1}\int \rho {\left|u\right|}^{2}dxdt,$$
where
(16)$$S=\{\left(\rho ,u\right)\;|\;\rho_{t}+\nabla \cdot \left(\rho u\right)=0,\rho \left(x,0\right)=\rho_{1},\rho \left(x,1\right)=\rho_{2}\}$$
is the admissible set.

The Wasserstein gradient flows are Eulerian descriptions to generalized diffusions [33]. Other choices of dissipation can define other metrics in the space of probability measures [34,35]. Within (Equation 15), the JKO scheme can be roughly approximated as
(17)$$\Phi^{k+1}=\underset{\Phi \in Diff}{\arg\min}\frac{1}{2\tau }\int_{\Omega }{\left|\Phi \left(x\right)-x\right|}^{2}\rho^{k}\left(x\right)dx+F\left[\rho^{k}\tilde{\circ }\Phi^{-1}\left(x\right)\right]dx,$$
under a constant velocity assumption, where $\rho^{k}\tilde{\circ }\Phi^{-1}\left(x\right)=\rho^{k}\left(\Phi^{-1}\left(x\right)\right)/\det F\left(\Phi^{-1}\left(x\right)\right)$. The scheme (Equation 17) can be viewed as an implicit Euler discretization to the flow map Equation (Equation 10) in $\left(t^{k},t^{k+1}\right)$. Mathematically, it is interesting to establish the equivalence or discrepancy between (Equation 14) and (Equation 17).

### 2.2. Micro-Macro Model for Polymeric Fluids

One successful application of EnVarA is building thermodynamically consistent micro-macro models for many complex fluids [36,37]. Compared with the macroscopic continuum mechanics approach, micro–macro models couple the macroscopic hydrodynamic equations with a microscopic kinetic theory, which describes the origin of the macroscopic stress tensor [38,39].

In the simplest micro–macro models of complex fluids, the polymer molecules are modeled as an elastic dumbbell consisting of two “beads” joined by a one-dimensional spring [36,37]. The microscopic configuration of an elastic dumbbell is described by an end-to-end vector between the two beads, $q\in R^{d}$. Let f(x,q,t) be the number density distribution function of finding a molecule with end-to-end vector q at position x∈Ω at time *t*. To build a micro–macro model by EnVarA, in addition to the macroscopic flow map x(X,t) at the physical space, one also needs to introduce a flow map at the configurational space, denoted by q(X,Q,t), where Q are Lagrangian coordinates in the configurational space. For a given q(X,Q,t), the microscopic velocity V(x,q,t) can be defined as
(18)$$V\left(x\left(X,t\right),q\left(X,Q,t\right),t\right)=q_{t}\left(X,Q,t\right).$$ Due to the conservation of mass, the density distribution function f(x,q,t) satisfies
(19)$$\frac{d}{dt}\int_{\Omega }\int_{R^{d}}f\left(x,q,t\right)dqdx=0,$$
which leads to the kinematics
(20)$$\partial_{t}f+\nabla \cdot \left(fu\right)+\nabla_{q}\cdot \left(fV\right)=0$$
in Eulerian coordinates.

In the framework of EnVarA, the micro–macro system can be modeled through an energy–dissipation law
(21)$$\begin{array}{cc} \; & \frac{d}{dt}\int_{\Omega }\left(\frac{1}{2}\rho {\left|u\right|}^{2}+\lambda_{p}\int_{R^{d}}k_{B}Tf\ln f+\Psi fdq\right)dx\\ \; & \;=-\int_{\Omega }\left(\eta_{s}{\left|\nabla u\right|}^{2}+\int_{R^{d}}\frac{\lambda_{p}\zeta }{2}f{\left|V-\tilde{V}\right|}^{2}dq\right)dx, \end{array}$$
where ρ is the constant density of the fluid, $\lambda_{p}>0$ is a constant that represents the polymer density, $k_{B}$ is the Boltzmann constant, *T* is the absolute temperature, $\eta_{s}>0$ is the solvent viscosity, the constant ζ is related to the polymer relaxation time, Ψ=Ψ(q) is the microscopic elastic potential of the polymer molecules. For Hookean and FENE models, the elastic potential Ψ(q) is given by $\Psi \left(q\right)=\frac{1}{2}H{\left|q\right|}^{2}$ and $\Psi \left(q\right)=-\frac{HQ_{0}^{2}}{2}\ln\left(1-{\left(\frac{\left|q\right|}{Q_{0}}\right)}^{2}\right)$ respectively, where H>0 is the elastic constant and $Q_{0}$ is the maximum dumbbell extension in FENE models. The second term of the dissipation accounts for the micro–macro coupling with $\tilde{V}$ being the macroscopic induced velocity. According to the Cauchy–Born rule, q=FQ due to the macroscopic flow, which indicates
$$\tilde{V}=\frac{d}{dt}\left(FQ\right)=\left(\frac{d}{dt}F\right)Q=\left(\nabla uF\right)Q=\nabla uq.$$ From the energy–dissipation law (Equation 21), one can derive the dynamics of the system by performing EnVarA in both micro- and macro scales. First, we look at the dynamics at the macroscopic scale. Due to the “separation of scale” [7], the second term in the dissipation (Equation 21) vanishes when deriving the macroscopic force balance. Since detF=1, the action functional can be written as
(22)$$A\left(x\right)=\int_{0}^{T}\int_{\Omega_{0}}\left[\frac{1}{2}\rho {\left|x_{t}\right|}^{2}-\lambda_{p}\int_{R^{d}}k_{B}Tf_{0}\ln f_{0}+\Psi \left(FQ\right)f_{0}dQ\right]dXdt$$
in Lagrangian coordinates, where $f_{0}\left(X,Q\right)$ is the initial number distribution function, and $f\left(x,FQ,t\right)=f_{0}\left(X,Q\right)$ due to detF=1. By applying the LAP, i.e., taking the variation of A(x) with respect to x, we obtain
(23)$$\frac{\delta A}{\delta x}=-\rho \left(u_{t}+u\cdot \nabla u\right)+\lambda_{p}\nabla \cdot \left(\int_{R^{d}}f\nabla_{q}\Psi ⊗qdq\right)$$
in Eulerian coordinates. Indeed, consider a perturbation $x^{\epsilon }=x+\epsilon y,$ where $y\left(X,t\right)=\tilde{y}\left(x\left(X,t\right),t\right)$ satisfying $\tilde{y}\cdot n=0$. Then
$$\begin{array}{cc} \frac{d}{dt}A\left(x^{\epsilon }\right){|}_{\epsilon =0}=\int_{0}^{T}\int_{\Omega_{0}} & \left[-\rho x_{tt}\cdot y-\lambda_{p}\int f_{0}\nabla_{q}\Psi ⊗Q:\nabla_{X}ydQ\right]dXdt. \end{array}$$ Pushing forward to Eulerian coordinates, we have
$$\begin{array}{cc} \; & \frac{d}{dt}A\left(x^{\epsilon }\right){|}_{\epsilon =0}=\int_{0}^{T}\int_{\Omega }\left[-\rho \left(u_{t}+u\cdot \nabla u\right)\cdot \tilde{y}-\lambda_{p}\int f\nabla_{q}\Psi ⊗q:\nabla_{x}\tilde{y})dq\right]dxdt\\ \; & =\int_{0}^{T}\int_{\Omega }\left(-\rho \left(u_{t}+u\cdot \nabla u\right)+\lambda_{p}\nabla \cdot \left(\int f\nabla_{q}\Psi ⊗qdq\right)\right)\cdot \tilde{y}dxdt, \end{array}$$
which leads to (Equation 23). For the dissipation part, the MDP, i.e., taking the variation of D with respect to $x_{t}$, leads to
(24)$$\frac{\delta D}{\delta x_{t}}=-\eta_{s}\Delta u+\nabla p,$$
where *p* is the Lagrangian multiplier for the incompressible condition ∇·u=0. Hence, the macroscopic force balance results in the momentum equation
(25)$$\rho \left(u_{t}+u\cdot \nabla u\right)+\nabla p=\eta_{s}\Delta u+\nabla \cdot \tau ,$$
where
(26)$$\tau =\lambda_{p}\int_{R^{d}}f\nabla_{q}\Psi ⊗qdq$$
is the induced stress from the configuration space, representing the microscopic contributions to the macroscopic level.

On the microscopic scale, similar to the generalized diffusions discussed in the last subsection, by taking variations with respect to q(X,Q,t) and V(X,Q,t), we obtain
$$\frac{\zeta }{2}\left(V-\nabla uq\right)=-\nabla_{q}\left(k_{B}T\ln f+1+\Psi \right).$$ Combining with Equation (Equation 20), we obtain the equation on the microscopic scale:(27)$$f_{t}+\nabla \cdot \left(uf\right)+\nabla_{q}\cdot \left(\nabla uqf\right)=\frac{2}{\zeta }\nabla_{q}\cdot \left(f\nabla_{q}\Psi \right)+\frac{2k_{B}T}{\zeta }\Delta_{q}f.$$ In summary, the final micro–macro system reads as follows:(28)$$\left\{\begin{array}{cc} \; & \rho \left(u_{t}+u\cdot \nabla u\right)+\nabla p=\eta_{s}\Delta u+\nabla \cdot \tau ,\;\nabla \cdot u=0,\\ \; & \tau =\lambda_{p}\int_{R^{d}}f\nabla_{q}\Psi ⊗qdq,\\ \; & f_{t}+\nabla \cdot \left(uf\right)+\nabla_{q}\cdot \left(\nabla uqf\right)=\frac{2}{\zeta }\nabla_{q}\cdot \left(f\nabla_{q}\Psi \right)+\frac{2k_{B}T}{\zeta }\Delta_{q}f, \end{array}\right.$$
subject to a suitable boundary condition.

### 2.3. Kinetic Fokker–Planck Equation

Using a similar micro-macro approach, one can also formulate the kinetic Fokker–Planck equation into an EnVarA formulation. The kinetic Fokker–Planck equation, given by
(29)$$\partial_{t}f+\nabla_{x}\cdot \left(vf\right)+\nabla_{v}\cdot \left(\left(-\gamma v-\nabla \Phi \right)f\right)=\frac{\sigma^{2}}{2}\gamma \Delta_{v}f,$$
describes the evolution of the probability density function f(x,v,t) of a system of particles that satisfies a Langevin dynamics
(30)$$\left\{\begin{array}{cc} \; & dX_{t}=V_{t}dt\\ \; & dV_{t}=\left(-\nabla \Phi \left(X_{t}\right)-\gamma V_{t}\right)dt+\sigma dW_{t}, \end{array}\right.$$
where $W_{t}$ is the standard Wiener process and Φ(x) is the potential function. Due to the fluctuation–dissipation theorem (FDT), $\sigma =\sqrt{2k_{B}T\gamma }$, which ensures that the system admits an energy–dissipation law and can reach an equilibrium state [40].

The total energy of the system can be defined by
(31)$$E^{\mathrm{total}}=\int \int \frac{1}{2}f{\left|v\right|}^{2}+f\Phi +k_{B}Tf\ln fdxdv.$$ Then the kinetic Fokker–Planck equation can be written as two parts [41]
$$\partial_{t}f=L_{C}\left(x,v\right)f+L_{D}\left(x,v\right)f,$$
where
$$L_{C}=-\nabla_{x}\cdot \left(vf\right)+\nabla_{v}\cdot \left(\nabla \Phi f\right)$$
conserves the total energy $E^{\mathrm{total}}$, while
$$L_{D}=\nabla_{v}\cdot \left(\gamma fv+\frac{\sigma^{2}}{2}\nabla_{v}f\right)$$
dissipates $E^{\mathrm{total}}$. Indeed, a direct computation shows that
$$\begin{array}{cc} \; & \frac{d}{dt}\;\int \int \;\frac{1}{2}f{\left|v\right|}^{2}+f\Phi +k_{B}Tf\ln fdxdv=\;\int \int \;\left[\frac{1}{2}{\left|v\right|}^{2}+\Phi +k_{B}T\ln f\right]f_{t}dvdx\\ \; & \;=\;\int \int \;\left[\frac{1}{2}{\left|v\right|}^{2}+\Phi +k_{B}T\ln f\right]\left(L_{C}f+L_{D}f\right)dvdx. \end{array}$$ Since
$$\begin{array}{cc} \; & \;\int \int \;\left[\frac{1}{2}{\left|v\right|}^{2}+\Phi +k_{B}T\ln f\right]L_{C}fdvdx\\ \; & =\;\int \int \;\nabla_{x}\left[\frac{1}{2}{\left|v\right|}^{2}+\Phi +k_{B}T\ln f\right]\cdot vf-\nabla_{v}\left[\frac{1}{2}{\left|v\right|}^{2}+\Phi +k_{B}T\ln f\right]\cdot \nabla \Phi fdvdx\\ \; & =\;\int \int \;\nabla_{x}\Phi \cdot vf+k_{B}T\nabla_{x}\left(\ln f\right)\cdot vf-v\cdot \nabla \Phi f-k_{B}T\nabla_{v}\ln f\cdot \nabla \Phi fdvdx\\ \; & =\;\int \int \;k_{B}T\left(\nabla_{x}f\cdot v-\nabla_{v}f\cdot \nabla \Phi \right)dvdx=0, \end{array}$$
where the last equality follows the fact that $\nabla_{x}\cdot v=0$ and $\nabla_{v}\cdot \left(\nabla \Phi \right)=0$. Hence, $f_{t}=L_{C}f$ conserves the total energy. In the meantime,
$$\begin{array}{cc} \;\int \int \;\left[\frac{1}{2}{\left|v\right|}^{2}+\Phi +k_{B}T\ln f\right]L_{D}fdvdx= & \;\int \int \;\left(v+K_{B}T\nabla_{v}\ln f\right)\cdot \left(-f\gamma v-\frac{\sigma^{2}}{2}\nabla_{v}f\right))\\ \; & =\;\int \int \;-\gamma f|v+k_{B}T\nabla_{v}{\ln f|}^{2}dvdx\leq 0. \end{array}$$ Here, we need $\sigma^{2}=2k_{B}T\gamma$ to obtain the quadratic form. $\sigma =\sqrt{2k_{B}T\gamma }$ is known as the fluctuation–dissipation theorem [30]. This calculation also reveals the connection between the fluctuation–dissipation theorem and linear response theorem.

Similar to the micro–macro approach for polymeric fluids introduced in a previous subsection, one can treat v as a microscopic variable, and x as a macroscopic variable; the micro–macro coupling is imposed through $x_{t}=v$. Due to the conservation property, the probability density f(x,v,t) satisfies the kinematics
(32)$$f_{t}+\nabla_{x}\cdot \left(vf\right)+\nabla_{v}\cdot \left(U\left(x,v\right)f\right)=0,$$
where U can be decomposed as
(33)$$U=U_{C}\left(x\right)+U_{D}\left(v\right),$$ Here $U_{C}\left(x\right)$ is a conserve velocity. In other words, the equation $f_{t}+\nabla_{x}\cdot \left(vf\right)+\nabla_{v}\cdot \left(U_{C}\left(x\right)f\right)=0$ conserves the total energy, i.e.,
(34)$$\frac{d}{dt}\;\int \int \;\frac{1}{2}f{\left|v\right|}^{2}+f\Phi \left(x\right)+f\ln fdvdx=0.$$
if $f_{t}+\nabla_{x}\cdot \left(vf\right)+\nabla_{v}\cdot \left(U_{C}\left(x\right)f\right)=0$. Notice that
(35)$$\begin{array}{cc} \; & \;\int \int \;\left[\frac{1}{2}{\left|v\right|}^{2}+\Phi +k_{B}T\ln f\right]\left(L_{C}fdv\right)dx\\ \; & =\;\int \int \;\nabla_{x}\Phi \cdot vf-v\cdot U_{c}\left(x\right)f+k_{B}T\nabla_{x}f\cdot v-k_{B}T\nabla_{v}f\cdot \nabla \Phi dvdx\\ \; & =\;\int \int \;(\nabla_{x}\Phi -U_{C}\left(x\right))\cdot vdvfdx, \end{array}$$
which indicates that $U_{C}\left(x\right)=\nabla_{x}\Phi$. In other word, we can obtain $U_{C}\left(x\right)$ by applying the least action principle at the macroscopic scale.

The dissipative velocity $U_{D}\left(v\right)$ can be obtained through the energy–dissipation law
(36)$$\frac{d}{dt}{\int f|v|}^{2}+f\ln f+f\Phi dv=\int -\gamma f{\left|U_{D}\right|}^{2}dv.$$ A standard variational procedure results in
(37)$$\gamma fU_{D}\left(v\right)=fv+K_{B}T\nabla_{v}f.$$ Here $\frac{1}{2}f{\left|v\right|}^{2}$ plays a role of the internal energy in the macroscopic scale. The EnVarA formulation to the kinetic Fokker–Planck equation may open a new door for both theoretical and numerical study of these types of equations.

## 3. EnVarA for Chemical Reactions

Chemical reactions play important roles in many physical, chemical, and biological processes [42]. The fundamental “law” for chemical reaction kinetics is the law of mass action (LMA), which is discovered by Waage and Guldberg, and H. van’t Hoff independently in the 19th century. A reversible chemical reaction system with *N* species $\left\{X_{1},X_{2},\ldots X_{N}\right\}$ and *M* reactions can often be represented by
$$\alpha_{1}^{l}X_{1}+\alpha_{2}^{l}X_{2}+\ldots \alpha_{N}^{l}X_{N}\overset{k_{l}^{+}}{\underset{k_{l}^{-}}{⇌}}\beta_{1}^{l}X_{1}+\beta_{2}^{l}X_{2}+\ldots \beta_{N}^{l}X_{N},\;l=1,\ldots ,M,$$
where $\alpha_{i}^{l}$ and $\beta_{i}^{l}$ are constant coefficients. Let $c={\left(c_{1},c_{2},\ldots ,c_{N}\right)}^{T}$ be concentrations of all species. Then c satisfies the reaction kinetics
(38)$$\partial_{t}c_{i}=\sum_{l=1}^{M}\sigma_{il}r_{l}\left(c\right),$$
where $r_{l}\left(c\right)$ is the reaction rate for the *l*-th chemical reaction, and $\sigma_{il}=\beta_{i}^{l}-\alpha_{i}^{l}$ is the stoichiometric coefficients. From (Equation 38), it is noticed that
(39)$$\frac{d}{dt}\left(e\cdot c\right)=e\cdot \sigma r\left(c\left(t\right),t\right)=0,\;for\;\;\;e\in Ker\left(\sigma^{T}\right).$$ In turn, one can define *N-rank*(σ) linearly independent conserved quantities for the reaction network.

The classical LMA states that the reaction rate is directly proportional to the product of the reactant concentrations, i.e.,
(40)$$r_{l}\left(c\right)=k_{l}^{+}c^{\alpha^{l}}-k_{l}^{-}c^{\beta^{l}},\;c^{\alpha^{l}}=\prod_{i=1}^{N}c_{i}^{\alpha_{i}^{l}},\;c^{\beta^{l}}=\prod_{i=1}^{N}c_{i}^{\beta_{i}^{l}},$$
in which $k_{l}^{+}$ and $k_{l}^{-}$ are the forward and backward reaction constants for the *l*-th reaction. The LMA was designed for the ideal gas of a perfect gas of non-interacting point particles without charge or diameter. To generalize the LMA to more complicated systems, it is important to establish a thermodynamics basis of this empirical law.

Since 1950s, there has been a large amount of work aiming to build an Onsager-type variational theory for systems involving chemical reactions [1,43,44,45,46,47,48,49,50,51,52].The key idea is to build on analogies between continuum mechanics and reaction kinetics [1,52,53]. Inspired by these prior works, in a recent paper [54], the authors extended the EnVarA formulation to a system that involves chemical reactions. By introducing the reaction trajectory $R\in R^{M}$, which accounts for the “number” of forward chemical reactions that has occurred by time *t*, the reaction kinetics can be reformulated in terms of R. The reaction trajectory R, which is known as the internal state variable in [43], is analogous to the flow map x(X,t) in mechanical systems [55]. The relation between species concentration $c\in R_{+}^{N}$ and the reaction trajectory R is given by $c=c_{0}+\sigma R,$ where $c_{0}$ is the initial concentration, and $\sigma \in R^{N\times M}$ is the stoichiometric matrix with $\sigma_{il}=\beta_{i}^{l}-\alpha_{i}^{l}$. The reaction rate r, defined as $\partial_{t}R$, is the reaction velocity [52]. Within the reaction trajectory, we can describe the chemical kinetics of a reaction network by an energy–dissipation law in terms of R and $\partial_{t}R$: (41)$$\frac{d}{dt}F\left[R\right]=-D_{\mathrm{chem}}\left[R,\partial_{t}R\right],$$
where $D_{\mathrm{chem}}\left[R,\partial_{t}R\right]$ is the rate of energy dissipation due to the chemical reaction.

Unlike mechanical processes, chemical reactions are usually far from equilibrium. The linear response assumption may be not valid unless at the last stage of chemical reactions [1,56]. As a consequence, $D_{\mathrm{chem}}$ is not quadratic in terms of $\partial_{t}R$ in general. For a general nonlinear dissipation
(42)$$D_{\mathrm{chem}}\left[R,\partial_{t}R\right]=\left(\Gamma \left(R,\partial_{t}R\right),\partial_{t}R\right)=\sum_{l=1}^{M}\Gamma_{l}(R,\partial_{t}R)\partial_{t}R_{l}\geq 0,$$
the reaction rate can be derived as [54,57]: (43)$$\Gamma \left(R,\partial_{t}R\right)=-\frac{\delta F}{\delta R},$$
which is the “force balance” equation for the chemical part [54,57]. It is often assumed that $\Gamma_{l}\left(R,\partial_{t}R\right)=\Gamma_{l}\left(R_{l},\partial_{t}R_{l}\right)$. Equation (Equation 43) specifies the reaction rate of the *l*-th chemical reaction. In this formulation, the free energy determines the chemical equilibrium, while the dissipation functional $D_{\mathrm{chem}}\left[R,\partial_{t}R\right]$ determines the reaction rate. It is worth mentioning that the variational principle (Equation 43) can be applied to other systems with non-quadratic dissipations, such as the systems studied in [58,59].

The classical LMA can be obtained by taking
(44)$$F\left[c\left(R\right)\right]=\sum_{i=1}^{N}c_{i}\ln\left(c_{i}/c_{i}^{\infty }\right),\;\Gamma_{l}\left(R,\partial_{t}R\right)=\ln\left(\frac{\partial_{t}R_{l}}{\eta_{l}\left(c\right)}+1\right),$$
where $c^{\infty }$ is an equilibrium of the reaction kinetics system that satisfies the detailed balance condition, i.e., $r_{l}\left(c^{\infty }\right)=k_{l}^{+}{\left(c^{\infty }\right)}^{\alpha^{l}}-k_{l}^{-}{\left(c^{\infty }\right)}^{\beta^{l}}=0$, and $\eta_{l}\left(c\right)=k_{l}^{-}\Pi_{i=1}^{N}c_{i}^{\beta_{i}^{l}}$. Similar to the cases in [58,59], near a chemical equilibrium, $\partial_{t}R_{l}$ is very small, and the dissipation of the LMA can be well approximated as a classical Onsager quadratic form of $\partial_{t}R_{l}$.

As an illustration, we consider a single chemical reaction
(45)$$\alpha_{1}X_{1}+\alpha_{2}X_{2}\overset{k_{l}^{+}}{\underset{k_{l}^{-}}{⇌}}\beta_{3}X_{3},$$
in which $\sigma ={\left(-\alpha_{1},-\alpha_{2},\beta_{3}\right)}^{T}$. According to the previous discussion, the variational approach produces
(46)$$\ln\left(\frac{\partial_{t}R}{k_{1}^{-}c_{3}^{\beta_{3}}}+1\right)=-\frac{\partial F}{\partial R},$$
where
$$\frac{\partial F}{\partial R}=\sum_{i=1}^{3}\sigma_{i}\mu_{i}=\sum_{i=1}^{3}\sigma_{i}(\ln\left(c_{i}/c_{i}^{\infty }\right).$$ From (Equation 46), we can obtain the law of mass action
(47)$$\dot{R}=k_{1}^{-}c_{3}^{\beta_{3}}\left(\frac{1}{K_{eq}}\frac{c_{1}^{\alpha_{1}}c_{2}^{\alpha_{2}}}{c_{3}^{\beta_{3}}}-1\right)=k_{1}^{+}c_{1}^{\alpha_{1}}c_{2}^{\alpha_{2}}-k_{1}^{-}c_{3}^{\beta_{3}},$$
where $K_{eq}=\frac{k_{1}^{+}}{k_{1}^{-}}=\frac{c_{3}^{\beta_{3}}}{{\left(c_{1}^{\infty }\right)}^{\alpha_{1}}{\left(c_{2}^{\infty }\right)}^{\alpha_{2}}}$.

The energetic variational formulation of chemical reactions opens a new door to model a general chemo-mechanical system in a unified variational way. One of the simplest chemo-mechanical systems is reaction–diffusion type system. For a reaction–diffusion system, the concentration of each species $c_{i}$ satisfies the kinematics: (48)$$\partial_{t}c_{i}+\nabla \cdot \left(c_{i}u_{i}\right)={\left(\sigma \partial_{t}R\right)}_{i},\;i=1,2,\ldots N,$$
where $u_{i}$ is the average velocity of each species due to its own diffusion, $R\in R^{M}$ represents various reaction trajectories involved in the system, with $\sigma \in R^{N\times M}$ being the stoichiometric matrix as defined earlier. Then the reaction–diffusion equation can be modeled through the energy–dissipation law [54]: (49)$$\frac{d}{dt}F\left[c_{i}\right]=-\int_{\Omega }\left[\sum_{i=1}^{N}\eta_{i}\left(c_{i}\right){\left|u_{i}\right|}^{2}+\sum_{l=1}^{M}\partial_{t}R_{l}\ln\left(\frac{\partial_{t}R_{l}}{\eta (c(R))}+1\right)\right]dx.$$ We can employ EnVarA to obtain equations for the reaction and diffusion parts, respectively, i.e., to obtain the “force balance equation” of the chemical and mechanical subsystems, which leads to
(50)$$\left\{\begin{array}{cc} \; & \eta_{i}\left(c_{i}\right)u_{i}=-c_{i}\nabla \mu_{i},\;i=1,2,\ldots N,\\ \; & \ln\left(\frac{\partial_{t}R_{l}}{\eta (c(R))}+1\right)=-\sum_{i=1}^{N}\sigma_{i}^{l}\mu_{i},\;l=1,\ldots ,M. \end{array}\right.$$ By taking $\eta_{i}\left(c_{i}\right)=\frac{1}{D_{i}}c_{i}$, we have a reaction–diffusion system
(51)$$\partial_{t}c_{i}=D_{i}\Delta c_{i}+{\left(\sigma \partial_{t}R\right)}_{i}.$$ One special reaction–diffusion system is a one-species system, in which the concentration or density ρ satisfies the kinematics
(52)$$\rho_{t}+\nabla \cdot \left(\rho u\right)=\sigma \left(\rho \right)R_{t},$$
where u is the average velocity that describes the diffusion of the species, and ***R*** is the reaction trajectory that describes the birth and death process. We can model this type system by an energy–dissipation law
(53)$$\frac{d}{dt}F\left[\rho \right]=-\int M_{1}\left(\rho \right){\left|u\right|}^{2}+M_{2}\left(\rho \right)R^{2}dx.$$ These systems are related to the unbalanced optimal transport in machine learning [60,61,62,63,64]. Different systems can be derived by choosing the free energy and the dissipation differently, including the Kantorovich–Fisher–Rao gradient flows [65].

### 3.1. Dynamical Boundary Condition

For a better description of short-range interactions between the material and the boundary, PDE systems with dynamical boundary conditions have drawn lots of attention recently [66,67]. These type of systems have a wide application in studying moving contact lines [68], batteries [69], the integrate-and-fire model for neuron networks [70], and sticky Brownian motion [71], etc.

Consider a bounded domain Ω, and denote the densities in Ω as ρ and the density on Γ=∂Ω as σ. In the models with dynamical boundary conditions, the conservation of total mass indicates that
(54)$$\frac{d}{dt}\left(\int_{\Omega }\rho dx+\int_{\Gamma }\sigma dS\right)=0.$$ Hence, ρ and σ satisfy the kinematics in Eulerian coordinates
(55)$$\begin{array}{cc} \; & \rho_{t}+\nabla \cdot \left(\rho u\right)=0,\;\;x\in \Omega ,\\ \; & \rho u\cdot \nu =R_{t},\;\sigma_{t}+\nabla_{\Gamma }\cdot \left(\sigma v_{\Gamma }\right)=R_{t},\;\;x\in \Gamma , \end{array}$$
where ν is the outer normal of Ω and ***R*** is the reaction trajectory for the chemical reaction ρ⇌σ that represents the density exchange between the bulk and surface.

In general, a system with a dynamical boundary condition can be modeled through an energy–dissipation law
(56)$$\frac{d}{dt}\left(F_{b}\left(\rho \right)+F_{s}\left(\sigma \right)\right)=-\left(\int_{\Omega }\eta_{b}{\left(\rho \right)|u|}^{2}dx+\int_{\partial \Omega }\eta_{s}\left(\sigma \right){\left|v\right|}^{2}+R_{t}\Psi \left(R,R_{t}\right)dS\right)$$
where $F_{b}\left(\rho \right)$ and $F_{s}\left(\sigma \right)$ are free energies in the bulk and surface, respectively. The standard variational procedures lead to the force balance equations for the mechanical and chemical parts
(57)$$\left\{\begin{array}{cc} \; & \eta_{b}\left(\rho \right)u=-\rho \nabla \mu_{b}\left(\rho \right)\\ \; & \eta_{s}\left(\sigma \right)v=-\sigma \nabla_{\Gamma }\mu_{s}\left(\sigma \right)\\ \; & \Psi \left(R,R_{t}\right)=-\left(\mu_{s}\left(\sigma \right)-\mu_{b}\left(\rho \right)\right), \end{array}\right.$$
where $\mu_{s}\left(\sigma \right)-\mu_{b}\left(\rho \right)$ is the affinity of the bulk–surface reaction. Different choices of the free energy and the dissipation lead to different systems.

A typical example is the Cahn–Hilliard equation with dynamic boundary condition, which can be derived by taking
$$\begin{array}{cc} \; & F_{b}\left(\varphi \right)=\int_{\Omega }\frac{1}{2}{\left|\nabla \varphi \right|}^{2}+F\left(\varphi \right)dx,\;F_{s}\left(\psi \right)=\int_{\Gamma }\frac{1}{2}{\left|\nabla_{\Gamma }\psi \right|}^{2}+G\left(\psi \right)dS,\\ \; & \;\eta_{b}\left(\varphi \right)=\frac{1}{m_{b}}\varphi^{2},\;\eta_{s}\left(\psi \right)=\frac{1}{m_{s}}\psi^{2},\;\Psi \left(R,R_{t}\right)=\gamma R_{t}^{2}. \end{array}$$ Here the linear response assumption is used for the bulk–surface exchange. The final equation can be written as [67],
(58)$$\left\{\begin{array}{cc} \; & \varphi_{t}=m_{b}\Delta \mu_{b},\;\mu_{b}=-\Delta \varphi +F^{\prime }\left(\varphi \right)\;x\in \Omega \\ \; & \psi_{t}=m_{\Gamma }\Delta_{\Gamma }\mu_{s}-\frac{1}{\gamma }\left(\mu_{s}\left(\psi \right)-\mu_{b}\left(\varphi \right)\right),\;\mu_{s}=-\Delta_{\Gamma }\psi +G^{\prime }\left(\psi \right)\;x\in \Gamma \\ \; & \frac{\partial \mu }{\partial \nu }=\frac{1}{\gamma }\left(\mu_{s}\left(\psi \right)-\mu_{b}\left(\varphi \right)\right),x\in \Gamma , \end{array}\right.$$
subject to a suitable initial condition. In the limit γ→0, the model is reduced to the model proposed in [66], while limit γ→∞ corresponds to the model in [72].

### 3.2. Boltzmann Equation

Another application of the EnVarA formulation for reaction kinetics is to reformulate the Boltzmann equation of ideal gas into a variational form, which was also explored in some previous works [73,74]. The key point is to view the collisions between particles as generalized chemical reactions. From a historical perspective, Maxwell and Boltzmann used an analogue of the law of mass action for collisions and discovered the principle of detailed balance [75].

Consider a Boltzmann equation
(59)$$\frac{\partial f}{\partial t}+v\cdot \nabla_{x}f=Q\left(f,f\right),\;t\geq 0,\;x\in R^{N},\;v\in R^{N},$$
where f(x,v,t) is the density for particles at the point x, having velocity v at time *t*, Q(f,f) is the collision term that can be written as
(60)$$Q\left(f,f\right)=\int w\left(v^{\prime },v_{*}^{\prime }|v,v_{*}\right)f^{\prime }f_{*}^{\prime }-w\left(v,v_{*}|v^{\prime },v_{*}^{\prime }\right)ff_{*}dv_{*}dv^{\prime }dv_{*}^{{}^{\prime }}.$$ Here $f=f\left(x,v,t\right),\;f_{*}=f\left(x,v_{*},t\right),f^{\prime }=f\left(x,v^{{}^{\prime }},t\right),f_{*}^{\prime }=f_{*}^{\prime }\left(x,v_{*}^{{}^{\prime }},t\right)$, and $w(v,v_{*}|v^{\prime },v_{*}^{\prime })\geq 0$ is the transition probability for two particles with velocity v and $v_{*}$ before a collision to have velocities $v^{\prime }$ and $v_{*}^{\prime }$ after the collision.

The collisions can be viewed as generalized chemical reactions. Since collisions conserve mass, momentum, and kinetic energy, Q(f,f) satisfies
(61)$$\int_{R^{3}}Q\left(f,f\right)dv=0,\;\int_{R^{3}}v_{j}Q\left(f,f\right)dv=0,\;\int_{R^{3}}{\left|v\right|}^{2}Q\left(f,f\right)dv=0.$$ Moreover, due to the conservation of momentum and kinetic energy
(62)$$v+v_{*}=v^{\prime }+v_{*}^{\prime }{,\;|v|}^{2}+|v_{*}{|}^{2}=|v^{\prime }{|}^{2}+{\left|v_{*}^{\prime }\right|}^{2},$$
one can assume that $w(v,v_{*}|v^{\prime },v_{*}^{\prime })$ only depends on $v-v_{*}$ and $\omega \in S^{2}$, where ω is a parameter that determines $v^{\prime }$ and $v_{*}^{\prime }$, i.e.,
(63)$$v^{\prime }=v-\left(v-v_{*},\omega \right)\omega ,\;v_{*}^{\prime }=v_{*}+\left(v-v_{*},\omega \right)\omega .$$ For given v and $v_{*}$, a fixed ω defines a “chemical reaction”
(64)$$\left(v,v_{*}\right)\overset{k_{l}}{\underset{k_{2}}{⇌}}\left(v^{\prime },v_{*}^{\prime }\right)$$
with $k_{1}=k_{2}=B\left(v-v_{*},\omega \right)$, where $B(v-v_{*},\omega )$ is the collision kernel. At the Maxwellians, $f^{\prime }f_{*}^{\prime }-ff_{*}$ vanishes identically, so the system satisfies the detailed balance condition.

Like the kinetic Fokker–Planck equation, the kinematics of f(x,v,t) can be written as the sum of the conservative and dissipative parts, i.e.,
(65)$$\partial_{t}f=L_{C}f+L_{D}f,$$
where $L_{C}f=-\nabla_{x}\cdot \left(vf\right)$ and $L_{D}f=Q\left(f,f\right)$. The total energy of the system can be defined as
(66)$$F\left[f\right]=\int \frac{1}{2}f{\left|v\right|}^{2}+f\ln fdvdx.$$ For the conservative part,
(67)$$\begin{array}{cc} \langle \frac{\delta F}{\delta f},L_{C}f\rangle & =\int \nabla_{x}\cdot (\frac{1}{2}{\left|v\right|}^{2}+\ln f+1)\cdot vfdvdx\\ \; & =\int \nabla_{x}f\cdot vdvdx=-\int f\left(\nabla_{x}\cdot v\right)dvdx=0 \end{array}$$ For the dissipative part, a direct computation shows that
(68)$$\langle \frac{\delta F}{\delta f},L_{D}f\rangle =-\int \frac{1}{4}B\left(v-v_{*},\omega \right)\left(ff_{*}-f^{\prime }f_{*}^{\prime }\right)\ln\left(\frac{ff_{*}}{f^{\prime }f_{*}^{\prime }}\right)dv^{*}d\omega dv\leq 0$$ Here, ${\int Q\left(f,f\right)|v|}^{2}dv=0$ and ∫Q(f,f)dv=0 are used.

We can reformulate the energy–dissipation law of the Boltzmann Equation (Equation 68) in terms of reaction trajectories $R(v,v_{*},\omega )$, by imposing the kinematics
(69)$$f\left(v\right)=f_{0}\left(v\right)-\int \int R\left(v,v_{*},\omega \right)d\omega dv_{*}.$$ The dissipation can be written in terms of ***R*** and $\partial_{t}R$, i.e.,
(70)$$\int \int \int \frac{1}{4}R_{t}\ln\left(\frac{R_{t}}{Bf^{\prime }f_{*}^{\prime }}+1\right)d\omega \;dv^{*}dv.$$ Notice that
(71)$$\begin{array}{cc} \frac{d}{dt}\int f\left(\ln f-1\right)dv & =-\int \int \int \ln f\partial_{t}R\left(v,v_{*},\omega \right)d\omega dv_{*}dv\\ \; & =\int \int \int \frac{1}{4}\left(-\ln f-\ln f_{*}+\ln f^{\prime }+\ln f_{*}^{\prime }\right)\partial_{t}Rd\omega dv_{*}dv. \end{array}$$ By a variational procedure, we can obtain
(72)$$\ln\left(\frac{B^{-1}R_{t}}{f^{\prime }f_{*}^{\prime }}+1\right)=\left(\ln f+\ln f_{*}-\ln f^{\prime }-\ln f_{*}^{\prime }\right),$$
which leads to
(73)$$R_{t}\left(v,v*,\omega \right)=B\left(v-v_{*},\omega \right)\left(ff_{*}-f^{\prime }f_{*}^{\prime }\right).$$ It is worth mentioning that $-\ln f-\ln f_{*}+\ln f^{\prime }+\ln f_{*}^{\prime }$ is the affinity of the chemical reaction (Equation 64). Indeed, the chemical potential associated with v can be computed as
(74)$$\mu =\frac{\delta F}{\delta f}=\ln f+1+\frac{1}{2}{\left|v\right|}^{2},$$
where $\frac{1}{2}{\left|v\right|}^{2}$ plays the role of the internal energy. Due to the conservation of the kinetic energy, we have ${\left|v\right|}^{2}+|v_{*}{|}^{2}=|v^{\prime }{|}^{2}+{\left|v_{*}^{\prime }\right|}^{2}$, which indicates that the affinity of the collision equals to $-\mu -\mu_{*}+\mu^{\prime }+\mu_{*}^{\prime }=-\ln f-\ln f_{*}+\ln f^{\prime }+\ln f_{*}^{\prime }$.

### 3.3. Reactive Fluids

Reactive and active fluids, which are systems involving coupling and competition between chemical reactions and various types of mechanical processes, have drawn lots of attention recently [76,77,78,79,80,81].

A simple example of reactive complex fluids is wormlike micellar (WLM) solutions, also known as “living polymer”. These are polymeric fluids that consist of long, cylindrical aggregates of self-assembled surfactants that can break and reform reversibly [82]. During the last couple of decades, a number of mathematical models have been proposed for wormlike micellar solutions [82,83,84,85,86]. The reptation–reaction model, proposed by Cates [82], is one of the first models that accounts for the reversible breaking and reforming of micellar chains. The Cates’ model assumes that a chain can break with a fixed probability per unit time per unit length anywhere along its length. Let c(L) be the number density of chains of length *L*, and the governing equation of c(L) can be formulated as
$$\begin{array}{cc} \dot{c}\left(L\right) & =-k_{1}Lc\left(L\right)-k_{2}c\left(L\right)\int_{0}^{\infty }c\left(L^{\prime }\right)dL^{\prime }+2k_{1}\int_{L}^{\infty }c\left(L^{\prime }\right)dL^{\prime }\\ \; & +k_{2}\int_{0}^{\infty }\int_{0}^{\infty }c\left(L^{\prime }\right)c\left(L^{\prime \prime }\right)\delta \left(L^{\prime }+L^{\prime \prime }-L\right)dL^{\prime }dL^{\prime \prime }, \end{array}$$
where $k_{1}$ and $k_{2}$ are rate constants for the breakage and recombination, respectively. It can be shown that the steady state ($\dot{c}\left(L\right)=0$) is given by $c^{\mathrm{eq}}\left(L\right)=\left(2k_{1}/k_{2}\right)\exp\left(-L/\bar{L}\right),$ where $\bar{L}$ is a constant that satisfies $2{\bar{L}}^{2}=\left(\int_{0}^{\infty }Lc\left(L\right)dL\right)k_{2}/k_{1}.$

Inspired by Cates’ seminal work, a two-species model for wormlike micellar solutions was proposed in [86], known as the Vasquez–Cook–McKinley (VCM) model. Although the VCM model was derived from a highly simplified discrete version of Cates’ model [87], it can capture the key rheological properties of wormlike micellar solutions [86,88,89]. However, due to the assumption that the breakage rate depends on the velocity gradient explicitly, the VCM model may not be thermodynamically consistent [87]. Using the generalized bracket approach [1], a thermodynamically consistent revision to the VCM model was proposed in [87,90], known as the GCB (Germann–Cook–Beris) model. They also extended such a model to a three species cases. Using the GENERIC framework [15,16], Grmela et al. also extended the VCM model into a three-species thermodynamically consistent model [84].

In a recent work [91], the authors developed a thermodynamically consistent two-species micro–macro model for wormlike micellar solutions, which incorporates a breakage and combination process of polymer chains into the classical micro–macro dumbbell model of polymeric fluids in the general framework of EnVarA. The model assumes that there exists only two species in the system. A molecule of species *A* can break into two molecules of species *B*, and two molecules of species *B* can reform species *A*. A polymer molecule of both species can be modeled as an elastic dumbbell consisting of two “beads” joined by a one-dimensional spring [37]. The microscopic configuration is described by an end-to-end vector $q\in R^{d}$. Let $\psi_{\alpha }\left(x,q,t\right)$ (α=A,B) be the number density distribution function of finding a molecule with end-to-end vector q at position x∈Ω at time *t* for species α.

In general, the breakage and combination processes can be regarded as chemical reactions
(75)$$q+q^{\prime }⇌q^{\prime \prime },$$
where q and $q^{\prime }$ are end-to-end vectors of species *B*, and $q^{\prime \prime }$ is an end-to-end of species *A* (see Figure 2a for illustration). We can denote the forward and backward reaction rates of (Equation 75) by $W^{+}\left(q,q^{\prime };q^{\prime \prime }\right)$ and $W^{-}\left(q,q^{\prime };q^{\prime \prime }\right)$, respectively. The kinematics of $\psi_{A}$ and $\psi_{B}$ can be written as
$$\left\{\begin{array}{cc} \; & \partial_{t}\psi_{A}+\nabla \cdot \left(u_{A}\psi_{A}\right)+\nabla_{q}\cdot \left(V_{A}\psi_{A}\right)=\int R_{t}\left(q^{\prime },q^{\prime \prime };q\right)dq^{\prime }dq^{\prime \prime }\\ \; & \partial_{t}\psi_{B}+\nabla \cdot \left(u_{B}\psi_{B}\right)+\nabla_{q}\cdot \left(V_{B}\psi_{B}\right)=-\int R_{t}\left(q,q^{\prime };q^{\prime \prime }\right)dq^{\prime }dq^{\prime \prime }-\int R_{t}\left(q^{\prime },q;q^{\prime \prime }\right)dq^{\prime }dq^{\prime \prime }\\ \; & R_{t}\left(q,q^{\prime };q^{\prime \prime }\right)=W^{+}\left(q,q^{\prime };q^{\prime \prime }\right)\psi_{B}\left(q\right)\psi_{B}\left(q^{\prime }\right)-W^{-}\left(q,q^{\prime };q^{\prime \prime }\right)\psi_{A}\left(q^{\prime \prime }\right), \end{array}\right.$$
where $u_{\alpha }$ and $V_{\alpha }$ are effective macroscopic and microscopic velocities. Different models can be obtained by choosing $W^{+}\left(q,q^{\prime };q^{\prime \prime }\right)$ and $W^{-}\left(q,q^{\prime };q^{\prime \prime }\right)$ differently. In the simplest case, one can take
(76)$$W^{\pm }\left(q,q^{\prime };q^{\prime \prime }\right)\neq 0\;\mathrm{if}\;\mathrm{and}\;\mathrm{only}\;\mathrm{if}\;q=q^{\prime }=q^{\prime \prime },$$
which corresponds to the case that an *A* molecule at position x with end-to-end vector q can only break into two *B* molecules with the same end-to-end vector, and the combination process can only happen between two *B* molecules at the same position x with the same end-to-end vector, as illustrated in Figure 2b, with α=1. Within this assumption, one can have a detailed balance condition for each x and q, and the kinematics can reduce to
(77)$$\left\{\begin{array}{cc} \; & \partial_{t}\psi_{A}+\nabla \cdot \left(u_{A}\psi_{A}\right)+\nabla_{q}\cdot \left(V_{A}\psi_{A}\right)=-R_{t}\\ \; & \partial_{t}\psi_{B}+\nabla \cdot \left(u_{B}\psi_{B}\right)+\nabla_{q}\cdot \left(V_{B}\psi_{B}\right)=2R_{t}, \end{array}\right.$$
where R(x,q,t) is the reaction trajectory for the breakage and combination for given q and x.

Within the above kinematics assumptions, the overall system can be modeled by an energy–dissipation law
(78)$$\begin{array}{cc} \; & \frac{d}{dt}\int_{\Omega }[\frac{1}{2}{\rho |u|}^{2}+\lambda \sum_{\alpha =1}^{2}\int \psi_{\alpha }\left(\ln\psi_{\alpha }-1\right)+\psi_{\alpha }U_{i}\left(q\right)dq]dx\\ \; & =-\int_{\Omega }{\left[\eta |\nabla u\right|}^{2}+\lambda \sum_{\alpha =1}^{2}\int \frac{\psi_{\alpha }}{\xi_{\alpha }}|V_{\alpha }-\nabla uq{|}^{2}+R_{t}\Gamma \left(R,R_{t}\right)dq]dx, \end{array}$$
where $\psi_{\alpha }\left(x,q,t\right)$ is the number density distributions of each species, $U_{i}\left(q\right)$ is the spring potential associated with each species, u is the macroscopic velocity satisfying the incompressible condition ∇·u=0, the constant ρ>0 is the density of the macroscopic flow, $V_{\alpha }$ is the average microscopic velocity of each species in the configuration space, λ>0 is the constant that represents the ratio between the kinetic energy and the elastic energy, and $\xi_{\alpha }>0$ is a constant related to the relaxation time of each species. By taking $\Gamma \left(R,R_{t}\right)=R_{t}\ln\left(\frac{R_{t}}{k_{2}\left(q\right)\psi_{B}^{2}}+1\right)$, the final micro–macro system can be derived as (see [91] for details)
(79)$$\left\{\begin{array}{cc} \; & \rho (\partial_{t}u+\left(u\cdot \nabla \right)u)+\nabla p=\eta \Delta u+\lambda \nabla \cdot \tau ,\\ \; & \nabla \cdot u=0,\\ \; & \partial_{t}\psi_{A}+u\cdot \nabla \psi_{A}+\nabla_{q}\cdot \left(\nabla uq\psi_{A}\right)-\xi_{A}\nabla_{q}\cdot \left(\nabla \psi_{A}+\nabla_{q}U_{A}\psi_{A}\right)=-\partial_{t}R,\\ \; & \partial_{t}\psi_{B}+u\cdot \nabla \psi_{B}+\nabla_{q}\cdot \left(\nabla uq\psi_{B}\right)-\xi_{B}\nabla_{q}\cdot \left(\nabla \psi_{B}+\nabla_{q}U_{B}\psi_{B}\right)=2\partial_{t}R, \end{array}\right.$$
where
(80)$$\partial_{t}R=k_{1}\left(q\right)\psi_{A}-k_{2}\left(q\right)\psi_{B}^{2},$$
and the stress tensor τ is given by
(81)$$\tau =\int \left(\nabla_{q}U_{A}⊗q\right)\psi_{A}+\left(\nabla_{q}U_{B}⊗q\right)\psi_{B}dq-\left(n_{A}+n_{B}\right)I$$
with $n_{\alpha }\left(x,t\right)=\int \psi_{\alpha }\left(x,q,t\right)dq$ being the number density of each species. The breakage and combination process indeed created an active stress in the macroscopic momentum equation. The global existence of the class solution near the global equilibrium of this model is given in [92]. The underlying variational structure (Equation 78) plays a crucial role in the theoretical analysis.

## 4. EnVarA for Non-Isothermal Systems

The above EnVarA framework works for isothermal cases, in which the whole system can be well-described by a single energy-dissipation law (Equation 1). However, in non-isothermal cases, one cannot obtain a single energy–dissipation law from the first and second laws of thermodynamics. Motivated by the classical treatments of temperature in rational mechanics [14,43], some recent work [93,94,95,96] extended EnVarA to non-isothermal cases.

To model a non-isothermal system, the extended EnVarA approach starts with a suitable form of the non-isothermal Helmholtz free energy density Ψ(ζ,ϑ), where ϑ(x,t) is the absolute temperature and ζ represents all the mechanical variables in the system, such as the density ρ(x,t) and the deformation tensor F, which are determined by the flow map x(X,t). The non-isothermal Helmholtz free energy density Ψ(ζ,ϑ) needs to be concave with respect to the temperature ϑ, i.e., $\Psi_{\vartheta \vartheta }<0$ [14].

According to the basic thermodynamical relations [14,97], the entropy density s=s(ζ,ϑ) is defined by
(82)$$s\left(\zeta ,\vartheta \right)=-\Psi_{\vartheta }\left(\zeta ,\vartheta \right),$$
while the internal energy density e(ζ,ϑ) is defined by
(83)$$e\left(\zeta ,\vartheta \right)=\Psi +s\vartheta =\Psi -\Psi_{\vartheta }\vartheta .$$ Since $\Psi_{\theta \theta }<0$, *s* is a monotonically increasing function of ϑ, which indicates that ϑ can be represented as a function of ζ and *s*, i.e.,
(84)ϑ=ϑ(ζ,s). As a consequence, the internal energy density *e* can be reformulated as a function of ζ and *s* by using (Equation 84). Without ambiguity, we denote $\hat{e}\left(\zeta ,s\left(\zeta ,\vartheta \right)\right)=e\left(\zeta ,\vartheta \right)$. According to the chain rule, it is straightforward to show
(85)$$\frac{\partial }{\partial s}\hat{e}\left(\zeta ,s\right)=\frac{\partial \Psi }{\partial \vartheta }\frac{\partial \vartheta }{\partial s}+\vartheta +s\frac{\partial \vartheta }{\partial s}=\vartheta ,$$
and
(86)$$\frac{\partial }{\partial \zeta_{i}}\hat{e}\left(\zeta ,s\right)=\frac{\partial \Psi }{\partial \zeta_{i}}+\frac{\partial \Psi }{\partial \vartheta }\frac{\partial \vartheta }{\partial \zeta_{i}}+s\frac{\partial \vartheta }{\partial \zeta_{i}}=\frac{\partial \Psi }{\partial \zeta_{i}}.$$ As in isothermal systems, it is crucial to specify the kinematics of the mechanical variable ζ as well as the kinematics of the temperature ϑ in non-isothermal systems. Different kinematic relations lead to different dynamics. To illustrate the idea, in the following, we only consider a simple case, with the mechanical variable being a density function ρ only, which satisfies the kinematics $\rho_{t}+\nabla \cdot \left(\rho u\right)=0$. For the temperature ϑ, we assume that
$$\vartheta_{t}+u\cdot \nabla \vartheta =0,$$
i.e., the temperature is purely transported along the trajectory. Moreover, we focus on cases where the kinetic energy K=0 throughout this section.

Although the whole system is no longer determined by the energy–dissipation law (Equation 1), we can still define the total energy $E^{\mathrm{total}}=\int \Psi \left(\rho ,\vartheta \right)dx$ as well as the dissipation potential D≥0, and derive the force balance Equation (Equation 4) for the mechanical part of the system by using the LAP and MDP. To simplify the calculations, we assume that
$$D=\frac{1}{2}\int \eta \left(\rho \right){\left|u\right|}^{2}dx$$
throughout this section, although other forms of the dissipation can be handled in a similar manner. By applying the LAP and the MDP, we can obtain the force balance equation for the mechanical part, i.e., equation for the flow map x(X,t). The computation procedure is almost the same as that in Section 2.1. Indeed, a direct computation shows that
$$\begin{array}{cc} \; & \delta A=-\delta \int_{0}^{T}\int_{\Omega_{0}}\Psi \left(\rho_{0}\left(X\right)/\det F,\vartheta_{0}\left(X\right)\right)\det F\;dXdt\\ \; & \;=-\int_{0}^{T}\int_{\Omega_{0}}-\left(\frac{\partial \Psi }{\partial \rho }\left(\frac{\rho_{0}\left(X\right)}{\det F},\vartheta_{0}\left(X\right)\right)\cdot \frac{\rho_{0}\left(X\right)}{\det F}+\Psi \left(\frac{\rho_{0}\left(X\right)}{\det F},\vartheta_{0}\left(X\right)\right)\right)\\ \; & \;\;\;\;\;\;\times (F^{-T}:\nabla_{X}\delta x)\det F\;\;dXdt. \end{array}$$ Pushing forward to the Eulerian coordinates, we have
(87)$$\begin{array}{cc} \delta A & =-\int_{0}^{T}\int_{\Omega }\left(-\mu \rho +\Psi \left(\rho ,\vartheta \right)\right)\nabla \cdot \left(\delta x\right)dxdt\\ \; & =\int_{0}^{T}\int_{\Omega }\left(-\rho \nabla \mu -s\nabla \vartheta \right)\cdot \delta xdxdt, \end{array}$$
which indicates that $\frac{\delta A}{\delta x}=-\rho \nabla \mu -s\nabla \vartheta$. Then the force balance equation for the mechanical part can be written as
(88)η(ρ)u=−ρ∇μ−s∇ϑ. To determine the equation for temperature ϑ or entropy *s*, we need to use both the first and second laws of thermodynamics. The first law of the thermodynamics can be written as
(89)$$\frac{d}{dt}\int_{V}\hat{e}\left(\rho ,s\right)dx=\int_{V}\nabla \cdot \Sigma +\nabla \cdot qdx,$$
where *V* is an arbitrary control volume, Σ stands for the specific work done to the system, and q is the absorbed heat flux. According to (Equation 85) and (Equation 86), it is easy to compute that
(90)$$\begin{array}{cc} \; & \frac{d}{dt}\int_{V}\hat{e}\left(\rho ,s\right)dx=\int_{V}\Psi_{\rho }\rho_{t}+\vartheta s_{t}dx\\ \; & =\int_{V}\Psi_{\rho }\left(-\nabla \cdot \left(\rho u\right)\right)+\vartheta \left(-\nabla \cdot \left(su\right)+\nabla \cdot \left(\frac{q}{\vartheta }\right)+{▵}^{*}\right)dx\\ \; & =\int_{V}-\nabla \cdot \left(\Psi_{\rho }\rho u+\vartheta su\right)+\rho \nabla \Psi_{\rho }\cdot u+s\nabla \vartheta u+\nabla \cdot q-\nabla \vartheta \cdot \left(\frac{q}{\vartheta }\right)+\vartheta {▵}^{*}dx, \end{array}$$
where the kinematic assumption on the entropy *s*
(91)$$\partial_{t}s+\nabla \cdot \left(su\right)=\nabla \cdot \left(\frac{q}{\vartheta }\right)+{▵}^{*},\;{▵}^{*}\geq 0$$
is used. Here, $\frac{q}{\vartheta }$ is the entropy flux defined by the Clausius–Duham relation, and ${▵}^{*}\geq 0$ is the density of the entropy production. Comparing (Equation 90) with (Equation 89), we obtain that
(92)$$\vartheta {▵}^{*}=-\rho \nabla \Psi_{\rho }\cdot u-s\nabla \vartheta \cdot u+\nabla \vartheta \cdot \left(\frac{q}{\vartheta }\right)$$ Recall (Equation 88). We have
(93)$$-\rho \nabla \Psi_{\rho }\cdot u-s\nabla \vartheta \cdot u=\eta \left(\rho \right){\left|u\right|}^{2}\geq 0.$$ On the other hand, by the Fourier law, we have
(94)$$q=k_{3}\nabla \vartheta ,$$
where $k_{3}\geq 0$. Hence, the density of the entropy production is
(95)$${▵}^{*}=\frac{{\rho |u|}^{2}}{\vartheta }+k_{3}\frac{{\left|\nabla \vartheta \right|}^{2}}{\vartheta^{2}}\geq 0,$$
which is consistent with the second law of thermodynamics.

The final system can be written as
(96)$$\left\{\begin{array}{cc} \; & \rho_{t}+\nabla \cdot \left(\rho u\right)=0\\ \; & \vartheta_{t}+u\cdot \nabla \vartheta =0\\ \; & s_{t}+\nabla \cdot \left(su\right)=\nabla \cdot \left(\frac{k_{3}\nabla \vartheta }{\vartheta }\right)+{▵}^{*}\\ \; & \eta \left(\rho \right)u=-\rho \nabla \mu -s\nabla \vartheta ,\;\mu =\frac{\partial \Psi }{\partial \rho }\\ \; & {▵}^{*}=\frac{{\rho |u|}^{2}}{\vartheta }+k_{3}\frac{{\left|\nabla \vartheta \right|}^{2}}{\vartheta^{2}}. \end{array}\right.$$ For ideal gas models, it is often assumed that
(97)$$\Psi \left(\rho ,\vartheta \right)=k_{1}\vartheta \rho \ln\rho -k_{2}\rho \vartheta \ln\vartheta ,\;\eta \left(\rho \right)=\rho .$$ Then
(98)$$s\left(\rho ,\vartheta \right)=-\left(k_{1}\rho \ln\rho -k_{2}\rho \left(\ln\vartheta +1\right)\right),\;e\left(\rho ,\vartheta \right)=k_{2}\rho \vartheta ,\;\rho u=-k_{1}\nabla \left(\rho \vartheta \right).$$ In this case, the non-isothermal system (Equation 96) can be simplified as
(99)$$\left\{\begin{array}{cc} \; & \rho_{t}=\nabla \cdot \left(k_{1}\nabla \left(\rho \vartheta \right)\right)\\ \; & {\left(k_{2}\rho \vartheta \right)}_{t}=\nabla \cdot \left(k_{1}\left(k_{1}+k_{2}\right)\vartheta \nabla \left(\rho \vartheta \right)+k_{3}\nabla \vartheta \right), \end{array}\right.$$
where the last equation follows (Equation 90).

One can derive a different system by imposing a different kinematics assumption on the temperature and the entropy. For instance, we can assume that the temperature ϑ is independent of the flow map x(X,t) during the purely mechanics processes, which leads to a simple kinematics $\partial_{t}\vartheta =0$. The force balance Equation (Equation 88) becomes η(ρ)u=−ρ∇μ. In this case, the kinematics of the entropy becomes
(100)$$\partial_{t}s=\nabla \cdot \left(\frac{q}{\vartheta }\right)+{▵}^{*},\;{▵}^{*}\geq 0.$$ Again, ${▵}^{*}$ is the density of the entropy production, and $\frac{q}{\vartheta }$ is the entropy flux defined by Clausius–Duhem relation. Following the same procedure, one can show that the entropy production is same as in (Equation 95). We refer the interested reader to [98] for a detailed derivation.

The formulations (Equation 91) and (Equation 100) are the differential version of the entropy balance equation [52,99]
(101)$$\frac{d}{dt}S=\frac{d_{e}}{dt}S+\frac{d_{i}}{dt}S,$$
introduced by Prigogine in [100] as a modern formulation of the second law of thermodynamics (See 3.4 in [52]). For mechanical processes, $S=\int_{V}s\left(x,t\right)dx$ is the entropy in the control volume *V*, $\frac{d_{e}}{dt}S=\int_{\partial V}J_{e}dS$ is the entropy change due to the exchange of matter and energy with $J_{e}$ being the entropy flux [52], while $\frac{d_{i}}{dt}S=\int_{V}{▵}^{*}dx\geq 0$ is the entropy change due to irreversible process, i.e., the entropy production. The Clausius–Duhem relation specifies $J_{e}=\frac{q}{\vartheta }$.

For systems involving chemical reactions, since the system is not determined by the flow map x(X,t) alone, it is natural to assume that the temperature also depends on the reaction trajectory R [43]. The remaining derivation is almost same as for the mechanical system.

## 5. Coarse-Graining and Numerical Realization

In this section, we briefly review some applications of the EnVarA in developing structure-preserving numerical schemes [28,57,101] and thermodynamically consistent coarse-grained models [91].

One of the fundamental questions in developing numerical schemes is how to preserve the properties of the original systems as much as possible. Different mathematical representations of physical principles can lead to different structure-preserving methods [102]. For dissipative systems modeled by the EnVarA, the most important structures are the underlying variational structures and the kinematic constraints of physical variables, such as positivity and the conservation property of a mass density, and the conservation of current for a charged system.

In [28], the authors introduced a numerical framework called the discrete energetic variational approach to preserve the energetic variational structure in the semi-discrete level. The approach first seeks a finite dimensional realization to a continuous energy–dissipation law, given by
(102)$$\frac{d}{dt}E_{h}\left(\Xi (t)\right)=-{▵}_{h}\left(\Xi \left(t\right),\Xi^{\prime }\left(t\right)\right),$$
where $\Xi \left(t\right)={\left(\Xi_{1}\left(t\right),\Xi_{2}\left(t\right),\ldots ,\Xi_{K}\left(t\right)\right)}^{\prime }\in R^{K}$ is the discrete state variable, $E_{h}\left(\Xi \right)$ is the discrete total energy that is the sum of the discrete kinetic energy $K_{h}\left(\Xi ,\Xi^{\prime }\right)$ and the discrete free energy $F_{h}\left(\Xi \right)$, and ${▵}_{h}\left(\Xi \left(t\right),\Xi^{\prime }\left(t\right)\right)=2D_{h}$ is the discrete energy dissipation. The discrete energy–dissipation (Equation 102) law can be obtained by introducing a suitable spatial discretization to the original system by Eulerian [103], Lagrangian [28,101], or particle approximations [104]. From the discrete energy–dissipation law (Equation 102), one can obtain an ODE system of Ξ(t), given by
(103)$$\frac{\delta D_{h}}{\delta \Xi^{\prime }}=\frac{\delta A_{h}}{\delta \Xi },$$
by the energetic variational approach. Here, $A_{h}\left(\Xi \left(t\right)\right)=\int_{0}^{T}K_{h}\left(\Xi^{\prime }\right)-F_{h}\left(\Xi \left(t\right)\right)dt$ is the discrete action functional. In the case that $K_{h}=0$ and $D_{h}\left(\Xi \left(t\right),\Xi^{\prime }\left(t\right)\right)$ is a quadratic function of $\Xi^{\prime }$, the ODE system of Ξ(t) can be written as
(104)$$M\left(\Xi \right)\Xi^{\prime }\left(t\right)=-\frac{\delta F_{h}}{\delta \Xi }\left(\Xi \left(t\right)\right),$$
where $M\left(\Xi (t)\right)$ is a K×K matrix.

The discrete energetic variational approach follows the idea of “discretize-then-variation”, which has been used to study both Hamiltonian and dissipative systems [18,105,106,107,108,109,110]. An advantage of this Ritz-type variational approach is that the resulting system inherits the variational structure from the continuous level, which enables us to apply these variational temporal discretizations [111,112,113]. For example, an implicit Euler discretization to Equation (Equation 104) could be reformulated as
(105)$$\begin{array}{cc} \; & \Xi =\underset{\Xi \in S_{ad}^{h}}{\arg\min}J_{n}\left(\Xi \right),\;J_{n}\left(\Xi \right)=\frac{\left(M_{*}^{n}\left(\Xi -\Xi^{n}\right),\left(\Xi -\Xi^{n}\right)\right)}{2\tau }+F_{h}\left(\Xi \right), \end{array}$$
where $S_{ad}^{h}$ is the admissible set of Ξ and $M_{*}^{n}=M\left(\Xi^{*}\right)$ is the numerical mobility that is independent with $\Xi^{n+1}$. The scheme is also known as the minimizing movement scheme [114,115]. Although in general, the optimization problem (Equation 105) is non-convex in a non-convex admissible set $S_{ad}^{h}$, one can use some line search-based optimization method to update $\Xi^{n+1}\in S_{ad}^{h}$ such that $F_{h}\left(\Xi^{n+1}\right)\leq F_{h}\left(\Xi^{n}\right)$. The existence of a minimizer of $J_{n}\left(\Xi \right)$ that decreases the total energy and the convergence of the numerical scheme at the discrete level can be proved under suitable conditions on $F_{h}\left(\Xi \right)$ [101,112,116]. A smaller τ can be chosen such that the optimization problem can be solved by some standard optimization method.

Similarly, the BDF2 scheme or a modified Crank–Nicolson scheme can also be reformulated as a minimization problem due to the variational structure in the semi-discrete level [113,117]. Other temporal discretization techniques, such as convex splitting [118,119,120,121], invariant energy quadratization (IEQ) [122,123], scalar auxiliary variable (SAV) [124,125], the discrete variational derivative method [105,126], and the strong stability-preserving (SSP) time discretizations [127] can also be applied to the semi-discrete system. In some applications, if the goal is to compute equilibrium states and the phase diagram, one can use some optimization method, such as the L-BFGS, to minimize the discrete free energy $F_{h}\left(\Xi \right)$ directly. Using the approach, we studied various confined liquid crystal systems [103,128,129,130].

The idea of “approximation-then-variation” can also be used to develop a dynamic coarse-grained model from a detailed multiscale model [18]. Similar to numerical approximations, the aim of coarse-graining is to reduce the degrees of freedom of the original system but preserve the properties of the original systems as much as possible. Closure approximation is one typical approach to develop coarse-grained models. The idea of closure approximation in to use the evolution equation of moments to represent the evolution of the probability density in the original Fokker–Planck type equations.

In [91], the authors studied maximum closure approximations to the micro–macro model of a wormlike micellar solution (Equation 79) by using “closure-then-variation”. The idea is to apply the closure approximation to the energy dissipation law first. Let A and B be the second moment of $\psi_{A}$ and $\psi_{B}$, and $n_{\alpha }$ be the number density of species α. By applying the energetic variational approach in the coarse-grained level, we can obtain a thermodynamically consistent closure model for both mechanical and chemical parts of the system. In more detail, within the assumption that $A=n_{A}\tilde{A}$ and $B=n_{B}\tilde{B}$, the free energy, obtained by the maximum entropy closure, can be reformulated in terms of number density $n_{A}$ and $n_{B}$, and the conformation tensor of two species $\tilde{A}$ and $\tilde{B}$, given by
(106)$$\begin{array}{cc} \; & {\tilde{F}}^{\mathrm{CL}}\left(n_{A},n_{B},\tilde{A},\tilde{B}\right)=\int n_{A}\left(\ln\left(\frac{n_{A}}{n_{A}^{\infty }}\right)-1\right)+n_{B}\left(\ln\left(\frac{n_{B}}{n_{B}^{\infty }}\right)-1\right)\\ \; & +\frac{n_{A}}{2}\left(-\ln\det\left(H_{A}\tilde{A}\right)+\mathrm{tr}\left(H_{A}\tilde{A}-I\right)\right)+\frac{n_{B}}{2}\left(-\ln\det\left(H_{B}\tilde{B}\right)+\mathrm{tr}\left(H_{B}\tilde{B}-I\right)\right)dx. \end{array}$$ We impose the kinematics for the number density to account for the macroscopic breakage and reforming procedure: (107)$$\left\{\begin{array}{cc} \; & \partial_{t}n_{A}+\nabla \cdot \left(n_{A}u\right)=-R^{n},\\ \; & \partial_{t}n_{B}+\nabla \cdot \left(n_{B}u\right)=2R^{n}, \end{array}\right.$$
where $R^{n}$ is the macroscopic reaction trajectory.

The dissipation of the moment closure system consists of three parts: the viscosity of the macroscopic flow, the evolution of the conformation tensors, and the reaction on the number density, which can be formulated as
(108)$$\begin{array}{cc} {\tilde{▵}}^{*}=\int & {\nu |\nabla u|}^{2}+\mathrm{tr}\left(M_{A}{\left(\frac{d\tilde{A}}{dt}\right)}^{2}\right)+\mathrm{tr}\left(M_{B}{\left(\frac{d\tilde{B}}{dt}\right)}^{2}\right)\;dx+{\tilde{D}}_{\mathrm{chem}}\left(R^{n},\partial_{t}R^{n}\right), \end{array}$$
where $\frac{d•}{dt}=\partial_{t}•+\left(u\cdot \nabla \right)•-\left(\nabla u\right)•-•{\left(\nabla u\right)}^{T}$ is the kinematic transport of the conformation tensor, $M_{A}\left(n_{A},\tilde{A}\right)$ and $M_{B}\left(n_{B},\tilde{B}\right)$ are mobility matrices. ${\tilde{D}}_{\mathrm{chem}}\left(R^{n},\partial_{t}R^{n}\right)$ is the dissipation for breakage and reforming at the macroscopic scale. A typical choice of ${\tilde{D}}_{\mathrm{chem}}\left(R^{n},\partial_{t}R^{n}\right)$ is
(109)$${\tilde{D}}_{\mathrm{chem}}\left(R^{n},\partial_{t}R^{n}\right)=\partial_{t}R^{n}\ln\left(\frac{\partial_{t}R^{n}}{\eta_{n}\left(R^{n}\right)}+1\right).$$ The choice of $\eta_{n}\left(R^{n}\right)$ determines the macroscopic reaction rate in the closure system. By taking
$$1/\eta^{n}\left(R^{n}\right)={\tilde{k}}_{2}\left(B\right)n_{B}^{2}\exp\left(\mathrm{tr}\left(\tau_{B}\right)/n_{B}\right)/\det\left(H_{B}\tilde{B}\right)$$
with ${\tilde{k}}_{2}\left(B\right)=k_{2}n_{B}^{d/2}/(2^{d}{\left(\pi \right)}^{d/2}{\left(\det\left(B\right)\right)}^{1/2}$, and applying the energetic variational approach, we can obtain the final moment closure system
(110)$$\left\{\begin{array}{cc} \; & \rho \left(\partial_{t}u+\left(u\cdot \nabla \right)u\right)+\nabla p=\eta \Delta u+\lambda \nabla \cdot \left(H_{A}A+H_{B}B-\left(n_{A}+n_{B}\right)I\right),\\ \; & \nabla \cdot u=0,\\ \; & \partial_{t}n_{A}+\nabla \cdot \left(n_{A}u\right)=-k_{1}^{\mathrm{neq}}n_{A}+k_{2}^{\mathrm{neq}}n_{B}^{2},\\ \; & \partial_{t}n_{B}+\nabla \cdot \left(n_{B}u\right)=2k_{1}^{\mathrm{neq}}n_{A}-2k_{2}^{\mathrm{neq}}n_{B}^{2},\\ \; & \partial_{t}\tilde{A}+\left(u\cdot \nabla \right)\tilde{A}-\left(\nabla u\right)\tilde{A}-\tilde{A}{\left(\nabla u\right)}^{T}=2\xi_{A}\left(I-H_{A}\tilde{A}\right),\\ \; & \partial_{t}\tilde{B}+\left(u\cdot \nabla \right)\tilde{B}-\left(\nabla u\right)\tilde{B}-\tilde{B}{\left(\nabla u\right)}^{T}=2\xi_{B}\left(I-H_{B}\tilde{B}\right), \end{array}\right.$$
where $A=n_{A}\tilde{A}$, $B=n_{B}\tilde{B}$, $k_{1}^{\mathrm{neq}}=k_{1}^{eq}\frac{\exp(\frac{1}{2}\mathrm{tr}\left(\tau_{A}/n_{A}\right))}{\sqrt{\det(H_{A}\tilde{A})}}$ and $k_{2}^{\mathrm{neq}}={\tilde{k}}_{2}^{eq}\frac{\exp(\mathrm{tr}\left(\tau_{B}/n_{B}\right)}{\det(H_{B}\tilde{B})}$. One can view (Equation 110) as a dynamics restricted in the sub–manifold formed by quasi–equilibrium states, in which $A=n_{A}\tilde{A}$ and $B=n_{B}\tilde{B}$. It is clear that the fluctuations on the number density $n_{A}$ and $n_{B}$ will create an active stress tensor in the macroscopic momentum equation. The resulting closure system is similar to the VCM [86] and GCB models [87]. Numerical simulations in [91] show that the momentum closure model can capture the key rheological features of wormlike micellar solutions as the VCM and GCB models.

## 6. Conclusions

In this paper, we briefly review some recent advances related to the energetic variational approaches, by examples. The general framework of EnVarA provides a thermodynamically consistent description to a complicated chemo-mechanical system involving couplings and competitions, such as the energetics vs. kinematics, the macroscopic hydrodynamics vs. micro-structures, the reversible vs. irreversible, the mechanics vs. chemistry, and the deterministic vs. stochastic. It is a powerful tool to study multiscale and multi-physics problem arising in physics, chemistry, and biology. The current review only sketches the underlying variational structure of several complex fluid models with chemo-mechanical coupling and non-isothermal effects; all these models require lots of investigation, both theoretically and numerically.

## Figures and Tables

**Figure 1 entropy-24-00721-f001:**
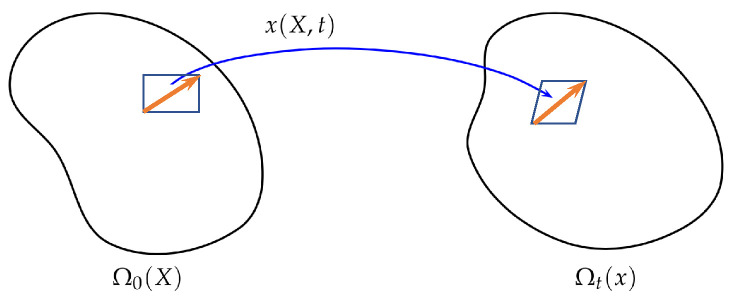
An illustration of the flow map.

**Figure 2 entropy-24-00721-f002:**
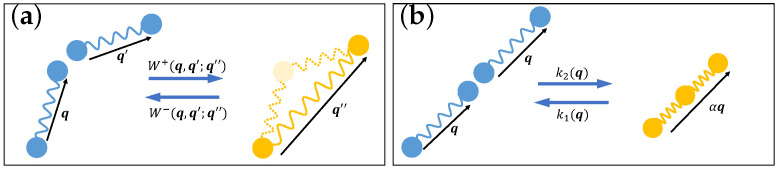
Schematic diagram of breakage and combination processes in wormlike micellar solutions, in which different species are indicated by different colors. (**a**) General reaction mechanism (Equation 75); (**b**) The reaction mechanism (Equation 76) considered in this paper (α=1).

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
