# Peer review of "Some Recent Advances in Energetic Variational Approaches"

_entropy, 2022, doi:10.3390/e24050721_

Round 1

Reviewer 1 Report

Referee Report for the manuscript “Recent Advances in Energetic Variational Approaches” by Yiwei Wang and Chun Liu

This manuscript deals with a complete description of the energetic variational approaches useful for building thermodynamically consistent models for complex fluids and other complex physical systems. This method is able to take into account several kinds of coupling between different physical quantities. In the paper, many examples have been worked out and carefully discussed: the application to continuum mechanics (including polymeric systems), kinetics of chemical reactions, non-isothermal systems, and applications to numerical schemes as well. The paper is interesting and well written; it is useful to disseminate this approach to the community working on non-equilibrium thermodynamics. The present reviewer suggests therefore the publication of this work on the Journal Entropy.  

I only ask to the authors to add a brief discussion concerning the following point. The classical Onsager theory, developed for small generalized forces, is based on linear phenomenological equations (between fluxes and forces) described by coefficients satisfying the Onsager reciprocity. Consistently, the entropy production is a quadratic form in the generalized forces. Recently, for an arbitrary nonequilibrium regime (namely, without limitations on the generalized forces), it has been proved that it is possible to obtain again the linear relation between fluxes and forces, described by symmetric Onsager coefficients. However, in the case with arbitrary forces, it has been proved that the rate of entropy production is nonlinear in thermodynamic forces thus giving a nonlinear generalization of the Onsager theory [1,2]. Only when the forces are sufficiently small, the rate of entropy production becomes equal to the classical Onsager quadratic form. The authors should explain if the energetic variational approaches are able to give similar results for arbitrary nonequilibrium regime and compare with results in Refs.[1,2].

[1] D. S. P. Salazar and G. T. Landi, Phys. Rev. Research 2,
033090 (2020).

[2] S. Giordano, Phys. Rev. E, 103, 052116 (2021).

Author Response

Point 1:  I only ask the authors to add a brief discussion concerning the following point. The classical Onsager theory, developed for small, generalized forces, is based on linear phenomenological equations (between fluxes and forces) described by coefficients satisfying the Onsager reciprocity. Consistently, entropy production is a quadratic form in the generalized forces. Recently, for an arbitrary nonequilibrium regime (namely, without limitations on the generalized forces), it has been proved that it is possible to obtain again the linear relation between fluxes and forces, described by symmetric Onsager coefficients. However, in the case of arbitrary forces, it has been proved that the rate of entropy production is nonlinear in thermodynamic forces thus giving a nonlinear generalization of the Onsager theory [1,2]. Only when the forces are sufficiently small, the rate of entropy production becomes equal to the classical Onsager quadratic form. The authors should explain if the energetic variational approaches are able to give similar results for arbitrary nonequilibrium regimes and compare with the results in Refs.[1,2].

[1] D. S. P. Salazar and G. T. Landi, Phys. Rev. Research 2,
033090 (2020).

[2] S. Giordano, Phys. Rev. E, 103, 052116 (2021).

Response 1: Thank you very much for your comments and for providing these two useful references. Indeed, we discuss how to deal with cases with non-quadratic dissipation in section 3, as reaction kinetics often have non-quadratic dissipation (see eq. (41) and (43)). We also related this with the references that you provided. Please see the colored text on pages 10 and 11.

Reviewer 2 Report

I uploaded a pdf file. 

Author Response

Thank you very much for your report. We sincerely apologize that the original submission contains too many typos and grammar errors. We have tried to carefully revise the entire manuscript (all typos that you pointed out in the report are fixed). We hope that you can give us more valuable comments on the revised manuscript. 

Reviewer 3 Report

The manuscript presents Onsager's variational principle and reviews its recent applications in fluid mechanics, chemical kinetics, and kinetic theory. The manuscript is well written. Both physical and mathematical aspects of the principle are well explained. Readers of Entropy will likely find the manuscript very informative and useful.